# Exploring the hospital patient journey: What does the patient experience?

**Raffaella Gualandi**[1]☯*, **Cristina Masella**[2]☯, **Daniela Viglione**[1], **Daniela Tartaglini**[1]

**1** Department of Nursing, Università Campus Bio-Medico di Roma, Rome, Italy, **2** Department of Management Economics and Industrial Engineering, Politecnico di Milano, Milan, Italy

☯ These authors contributed equally to this work.
* r.gualandi@unicampus.it

## Abstract

### Purpose

To understand how different methodologies of qualitative research are able to capture patient experience of the hospital journey.

### Methods

A qualitative study of orthopaedic patients admitted for hip and knee replacement surgery in a 250-bed university hospital was performed. Eight patients were shadowed from the time they entered the hospital to the time of transfer to rehabilitation. Four patients and sixteen professionals, including orthopaedists, head nurses, nurses and administrative staff, were interviewed.

### Results

Through analysis of the data collected four main themes emerged: the information gap; the covering patient-professionals relationship; the effectiveness of family closeness; and the micro-integration of hospital services. The three different standpoints (patient shadowing, health professionals' interviews and patients' interviews) allowed different issues to be captured in the various phases of the journey.

### Conclusions

Hospitals can significantly improve the quality of the service provided by exploring and understanding the individual patient journey. When dealing with a key cross-functional business process, the time-space dynamics of the activities performed have to be considered. Further research in the academic field can explore practical, methodological and ethical challenges more deeply in capturing the whole patient journey experience by using multiple methods and integrated tools.

**Data Availability Statement:** All relevant data are within the paper and its Supporting Information files.

**Funding:** The author(s) received no specific funding for this work.

**Competing interests:** The authors have declared that no competing interests exist.

## Introduction

In the healthcare knowledge-based system, literature has given increasing attention over time to improving clinical knowledge, including by making use of the patient's insider perspective [1–3]. In particular, patient experience of healthcare and the delivery of care is emerging as an important area of knowledge, but one that is sometimes overlooked [4, 5].

The Beryl Institute defines patient experience as "the sum of all interactions, shaped by an organization's culture, that influence patient perceptions, across the continuum of care"[6]. Wolf et al. specify that interactions are "The orchestrated touchpoints of people, processes, policies, communications, actions, and environment" and patient perceptions are "what is recognized, understood and remembered by patients and support people"[7].

In the last few years, emphasis on the emotional drivers of engagements has led many authors to enhance the customer experience starting from an analysis of the customer journey [8]. In the hospital context, the patient journey is a key cross-functional business process where patient and providers share action and information flows between people and systems across various touchpoints. Providers aim to manage hospital patient flow in order to provide safe and efficient patient care while ensuring the best use of hospital resources (i.e.: beds, operating theatres, clinics and specialized staff). Poor patient flow may result in decreasing levels of productivity, increasing risk of harming patients and decreasing levels of quality perceived by patients [9–11]. Patients aim to receive the best care together with a high quality of service. As a matter of fact, the patient is the only actor who experiences the whole path by connecting each step of the journey. Therefore, hospitals can significantly improve the quality of the service provided by exploring and understanding the individual patient journey [12–14].

Many tools may be used to measure and understand patient experience [15, 16]. Surveys are the methods mainly used to capture the patient experience and to evaluate the quality and safety of various clinical processes [17, 18]. However, questionnaires or traditional static observation may not be well-suited to reveal all the aspects of patient experience [19]. In the complex hospital environment, multiple factors can affect the patient experience, including the time-space dynamics of the activities performed and the patients' perceptions and emotions lived at the time of the experience [20]. Moreover, some authors emphasize that what the patient remembers is different from what he/she experiences in real-time, depending on the length of the recall period [21, 22]. Therefore, as what the patient remembers may change over time, gathering accurate and immediate data on the experience lived also depends on the time of the interview.

A recent study reports how the use of unstructured diaries completed in a patient's own words can capture the hospital-stay experience from the patient's own perspective. However, it is not clear how real-time experiences are reported in relation to high-emotional situations or clinical activities that can interfere with the patient's ability and willingness to write (i.e. during the transfer to the operating theatre or in the post-operative period, immediately after surgery). Furthermore, the authors show how study participants with a tertiary education wrote more in their diaries than those without [23]. This could potentially eliminate important aspects of the experience lived by vulnerable people.

Some authors have emphasized the value of shadowing for phenomenological research, by giving a more complete picture of the phenomenon in the real-time context of an organization [24, 25]. Patient shadowing may have an especially valuable role in gaining insights into complex cross-hospital processes, in particular when dealing with vulnerable people who could be excluded from interview studies [26, 27]. Furthermore, some studies have reported how, through shadowing methodology, it is possible to assess the lived experience of patients in a

patient-centred perspective [28, 29]. However, methodological and ethical issues of shadowing still need to be explored in greater depth [25, 30].

While on the one hand patient experience is increasingly considered as a driver for health services improvement, on the other it is still not clear how to capture the whole patient experience in traversing hospital services [31–33]. Therefore, this study seeks to explore which aspects of the hospital patient journey experience may be captured by the three different standpoints: patient shadowing, health professionals' interviews and patients interviews. Accordingly, it aims to answer the following questions: what does the patient experience through the hospital journey? How can it be captured?

## Materials and methods

### Study design

This study was a qualitative study with a phenomenological-hermeneutic approach using participant interviews and patient shadowing [34, 35]. The Consolidated Criteria for Reporting Qualitative Research—COREQ checklist was used as a guideline to report the study data [36]. The study was undertaken in a 250-bed Italian academic teaching hospital. Orthopaedic patients undergoing total hip (THA) or knee arthroplasty (TKA) were selected in order to analyse a standard clinical path (Fig 1). Urgently admitted patients were excluded due to the different clinical path they have to follow. The unit of analysis was the hospital patient's journey starting from the first outpatient visit and concluding with the first follow-up visit. The study was approved by the Hospital Ethics Committee.

### Data collection

Between August 2016 and April 2017, a total of twelve patients and sixteen key professionals were invited to participate and all agreed. There were no prior relationships between researchers and patients; two researchers knew some healthcare professionals because they worked in the same hospital, though in different units and without patient care roles. The possibility of coercion was minimized by guaranteeing data anonymity, by requesting voluntary participation in the study and by dealing with issues on which the researchers had no power to influence anything or anyone at hospital managerial level.

A convenience sample of patients was selected based on whether their inpatient admission and follow-up visit fell within the observation period. Inclusion criteria were: patients scheduled to undergo surgery for THA or TKA, for the shadowing phase; patients who had had a THA or TKA ad were in follow-up, for the interviews. Exclusion criteria were: <18 years, inability to understand, not wanting to participate, inability to read/speak Italian. Patients were asked to participate in the study at the time they arrived in the hospital. The first author invited patients to participate in the study when they met at the hospital for preoperative tests or on the day of admission. Patients accepting the invitation were provided with further information about the project by the first author, and were asked to sign their consent to participate in the study and to the anonymous use of their data.

Eight patients admitted between August and September 2016 were selected for the shadowing phase. A shadowing methodology was used in order to provide an embodied understanding of patients' experiences in context [26]. Two female students from the nursing and industrial engineering degree courses, with no roles in the delivery of patient care, were trained for data collection by the first author. In this way, the risk of not reporting negative feedback during the study by the participants, and subjective interpretations by the authors in capturing data, was minimized. Patients were shadowed from the time of hospital admission to the time of discharge, with the shadower observing the patient during daytime hours and completing a

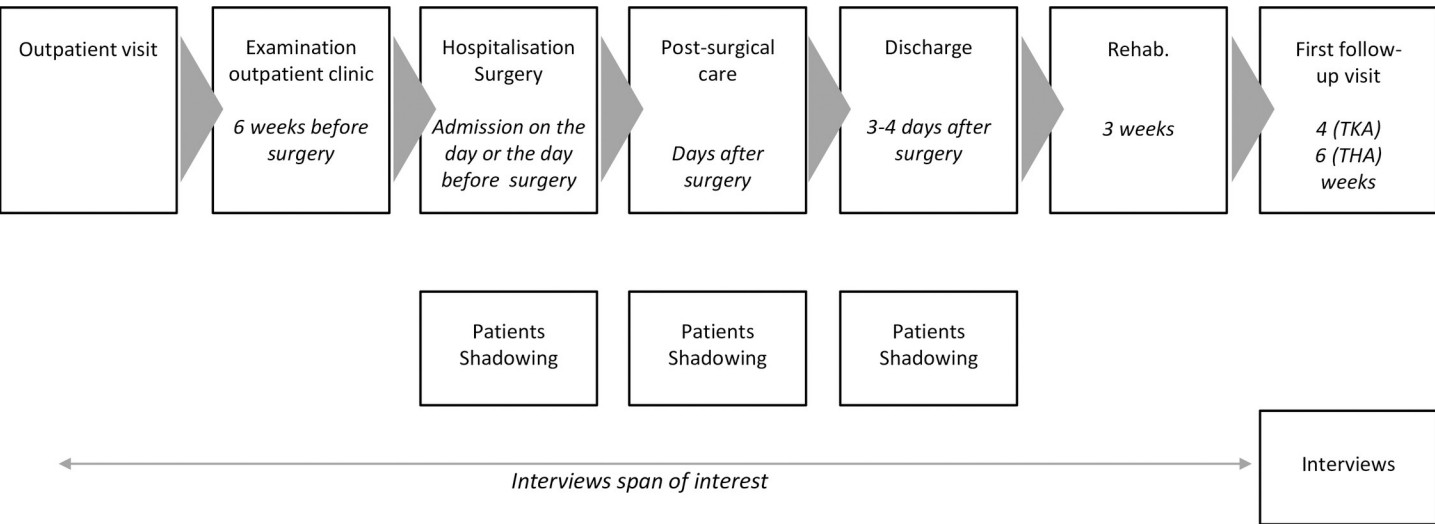

**Fig 1. Flow of patients' pathways in total hip arthroplasty (THA) / total knee arthroplasty (TKA) programme and scheduled timing of the study.**

data collection form prepared by the research group. This involved recording every step of the hospital journey process, by analysing each touchpoint and including time, patient, caregiver, activity, shadower's observations, and impressions. In particular, touchpoint observations indicated where patients and families go (setting), with whom they come into contact, how long the experience at each touchpoint takes (time), what patients and caregivers do, and a description of any comments of the patient and family, including any observable emotional state of the patient. By considering patients' emotion as consistent responses to internal or external events, the Plutchik's model was used as framework to understand its intensity in a positive or negative characterization [37]. In particular, Plutchik suggests emotions are low, medium or high-intensity, and if left unchecked, they can intensify. Accordingly, the patient's emotional journey was also assessed in reference to external events that altered the patient's emotional level.

Patient care procedures were not noted because they were not relevant for the current research objectives. The shadower observed the patients during all hospital transfers and entered the patients' rooms only to verify their general state and to gather any statements about their experience. The shadower was mainly passive during the observation, but was active in informal conversations. This level of proximity made it possible for the patients not to perceive shadowers as intrusive or disrespectful of their privacy.

Between September 2016 and April 2017, four patients and sixteen healthcare professionals participated in face-to-face open interviews lasting 30–45 minutes and performed by the first and the third author. A few main open questions were identified by the research group in order to analyse the main steps of the patients' journeys, the patients' experiences, and their reported emotions. Patients were interviewed at the first outpatient follow-up visit (Fig 1), scheduled one month after discharge from the ward, in order to include their perceptions of discharge.

In order to capture viewpoints representing various different roles, a collaborative purposive sampling technique was used among professionals with different level of professional experience who take care of orthopaedic patients. In particular, according to Benner's stages of clinical competence [38], two nurses with experience of at least five years, identified as expert nurses by their managers, and three nurses with experience of up to four years, identified as

competent nurses, were selected. In addition, two orthopaedic surgeons and one medical doctor under training were involved. Finally, three members of the administrative staff, the director responsible for the quality of care processes, and the head nurses of the units involved in the patient journeys (i.e.: two Ward Units, one Surgery Room, one Rehabilitation Unit), were interviewed.

All interviews were audio-recorded and transcribed verbatim with participant permission. Data from the field notes and the interviews were transferred to an Excel spreadsheet database to systematize them and for the subsequent analysis.

## Data analysis

Data analysis was performed on three levels as suggested by Ricoeur [30]: a naïve reading, a structural analysis and a critical analysis and discussion. The first author performed a thematic analysis of the text material. In the structural analysis, the units of meaning (what was said) were reflected in units of significance (what the texts were talking about) from which the key themes emerged (Table 1). Patients' emotions, reported or observed, were classified according to Plutchik's Wheel of Emotions [37]. After that, a critical analysis was carried out by the researchers in order to analyse the coding process, the categories and the meanings that emerged.

## Results

The main characteristics of the participants involved in the study are reported in Table 2. Patients involved ranged in age from 56–78 years with an average age of 67.3 years, and they were hospitalized on average 4.4 days. All had a regular clinical trajectory with no noteworthy complications. Healthcare professionals ranged in age from 29–61 years with an average age of 38.8 years and a work experience average of 10.6 years.

### The hospital patient journey

In the patient hospital journeys studied, seven main phases and forty-four consequent steps were traced by shadowing patients and interviewing the main actors. Table 3 shows which steps were identified from the interviews and which from the shadowing. In particular, the patient shadowing enabled more accurate reconstruction of all the steps, compared to what patients narrated after a period of time. This information can be obtained from the health professionals' interviews only by summarizing their different points of view. Furthermore, through shadowing it was possible to detect that within the hospital the patient went through eighteen different places and was in contact with more than fifty different health professionals. The patients' emotions as reported by the health professionals corresponded to what was

**Table 1. Illustration of structural analysis.**

| Units of meaning<br>What was said | Units of significance<br>What the text was talking about | Themes<br>Emergence of key themes |
|---|---|---|
| *"At the time of admission you can see from the face of the patient that he is shaken; they are not always aware; unfortunately, they do not always know perfectly what they are going to do"* (Healthcare professional 6). | Patient information does not seem to be effective. | The information gap |
| *"When I went to the operating theatre I met the doctor who was treating me. When he is there I am calm about what I have to do"* (Patient 2). | The presence of the surgeon in the operating theatre calms the patient. | The covering patient-professionals relationship |

**Table 2. Main characteristics of participant involved in the study.**

| **Patients** (interviewed) | | | | |
|---|---|---|---|---|
| **Code** | **Sex** | **Surgery*** | **LOS**, days** | **Clinical course** |
| Patient 1 | Female | TKA | 6 | Regular |
| Patient 2 | Female | TKA | 5 | Regular |
| Patient 3 | Female | TKA | 2 | Regular |
| Patient 4 | Male | THA | 6 | Regular |
| **Patients** (shadowed) | | | | |
| **Code** | **Sex** | **Surgery*** | **LOS**, days** | **Clinical course** |
| Patient 5 | Male | TKA | 6 | Regular |
| Patient 6 | Male | THA | 6 | Regular |
| Patient 7 | Female | THA | 6 | Regular |
| Patient 8 | Female | THA | 5 | Regular |
| Patient 9 | Male | TKA | 3 | Regular |
| Patient 10 | Male | THA | 4 | Regular |
| Patient 11 | Male | TKA | 2 | Regular |
| Patient 12 | Male | TKA | 2 | Regular |
| **Healthcare professionals** | | | | |
| **Code** | **Sex** | **Time from recruitment, years** | | |
| Healthcare professional 1 | Male | 20 | | |
| Healthcare professional 2 | Male | 1 | | |
| Healthcare professional 3 | Male | - | | |
| Healthcare professional 4 | Female | 4 | | |
| Healthcare professional 5 | Female | 4 | | |
| Healthcare professional 6 | Female | 7 | | |
| Healthcare professional 7 | Female | 16 | | |
| Healthcare professional 8 | Female | 5 | | |
| Healthcare professional 9 | Female | 3 | | |
| Healthcare professional 10 | Male | 3 | | |
| Healthcare professional 11 | Male | 10 | | |
| Healthcare professional 12 | Female | 20 | | |
| Healthcare professional 13 | Female | 15 | | |
| Healthcare professional 14 | Female | 15 | | |
| Healthcare professional 15 | Female | 17 | | |
| Healthcare professional 16 | Female | 20 | | |

* TKA = total knee arthroplasty; THA = total hip arthroplasty

**LOS = Length of Stay

observed by shadowing, but they did not match the general state of serenity reported by patients when interviewed.

The three different standpoints, (i.e.: patient shadowing, healthcare professional interviews and patient interviews) allowed different issues to be captured at the various phases of the journey. In particular, the shadowing was able to capture the 'connections' between one stage and another of the journey, such as movement from admissions to the ward and transport from the ward to the operating theatre, while the journey narrated by each professional and patient allowed the most significant touchpoints to be identified (Table 3).

When interviewed about a month after discharge, patients remembered a generally positive experience, linked specifically to the success of surgery and to a good relationship with the

**Table 3. Patient journeys' main steps and touchpoints.**

| Journeys' main steps and touchpoints | Touchpoints | Patients (Interviewed) | Patients (Shadowed) | Professionals (Interviewed) | Patient Main Emotions (shadowing and professionals interviews) |
|---|---|---|---|---|---|
| **Outpatient visit** | | | | | |
| Booking of the outpatient visit | ● | | - | ● | |
| Arrival at the hospital and administrative processing | ● | | - | ● | |
| Outpatient visit | ● | ● | - | ● | |
| Exit from the Hospital | | | - | | |
| **Examination at outpatient clinic** | | | | | |
| Call for pre-admission clinic | ● | | - | ● | Annoyance |
| Arrival at the hospital and waiting for procedures | ● | ● | - | ● | |
| Assistance procedures | ● | ● | - | ● | |
| Exit from the Hospital | | | - | ● | |
| **Hospitalization and surgery** | | | | | |
| Waiting for inpatient admission | | ● | - | ● | Fear, Apprehension |
| Call for inpatient admission notice and confirmation | ● | | - | ● | |
| Call for an informational meeting and evaluation of the therapy | ● | | - | ● | |
| Execution procedure for blood request | ● | | - | ● | |
| Informational meeting (when possible) | ● | | - | ● | |
| Arrival at the hospital and waiting for admission | ● | | ● | ● | |
| Administrative admission | ● | | ● | ● | |
| Moving to the ward | | | ● | ● | |
| Waiting in front of the Ward entrance | | | ● | ● | |
| Entry into the Ward | ● | ● | ● | ● | |
| Arrival at the inpatient room | ● | | ● | ● | |
| Waiting in the inpatient room | | | ● | ● | |
| Assistance procedures | ● | ● | ● | ● | |
| Transfer to the Operating Theatre | ● | ● | ● | ● | |
| Waiting in the Transfer bay | ● | | ● | ● | |
| Assistance procedures | ● | | ● | ● | |
| Entry into the Operating Theatre | ● | | ● | ● | |
| Transfer to the induction room | ● | | - | ● | |
| Surgery (unconscious patient) | | ● | - | ● | |
| Transfer to the post anaesthetic care unit (partially conscious patient) | ● | | - | ● | |
| **Post-surgical care** | | | | | |
| Transfer and entry to the Ward | ● | ● | ● | ● | Serenity, Fear, Apprehension |
| Assistance procedures | ● | ● | ● | ● | |
| Transfer and waiting for radiography | ● | | ● | ● | |
| Radiography | ● | ● | ● | ● | |
| Waiting for transfer | | | ● | | |
| Transfer to the Ward | ● | | ● | | |
| Post-surgery hospitalization | ● | ● | ● | ● | |
| **Discharge** | | | | | |
| Assistance procedures | ● | | ● | ● | Serenity |
| Transfer to the Rehabilitation Units | ● | ● | ● | ● | |

*(Continued)*

**Table 3.** (Continued)

| Journeys'main steps and touchpoints | Touchpoints | Patients (Interviewed) | Patients (Shadowed) | Professionals (Interviewed) | Patient Main Emotions (shadowing and professionals interviews) |
|---|---|---|---|---|---|
| **Rehabilitation stay** | | | | | |
| Arrival at the Rehabilitation Department | ● | | - | ● | Apprehension, Acceptance |
| Assistance procedures | ● | | - | ● | |
| Discharge | ● | | - | ● | |
| Rehabilitation in Outpatient | ● | | - | ● | |
| **Follow-up visit** | | | | | |
| Arrival at the hospital and administrative processing | ● | | - | ● | Serenity |
| Outpatient visit | ● | ● | - | ● | |
| Exit from the Hospital | | | - | | |

professionals. They showed appreciation and satisfaction and they declared that there were no major problems to deal with. One patient reported *"I was fine, look, I have to say the night of the surgery I was fine, the next day they also made me get up. They made me sit in the chair, my head was spinning a little, so it's not that ehm . . . then nothing else, everything else went well"* (Patient 1); Another reported *"What can I say? Better than that I don't think it is; that. . . we may be worse, but I have not found that I was worse, and I have only good things to say about the professor and all his assistants"* (Patient 3).

However, when shadowed, some discrepancies emerged. When going independently to the ward patients experienced confusion and anxiety, due to not having clearly understood indications, and to the waiting times before entering the assigned ward (Patient 5, 6, 7, 8, 9,12). Another critical step was the transfer and waiting in the operating theatre. They felt 'lost' when they were transferred and emotions of fear and anxiety emerged (Patient 10; 11). These experiences also emerged from the interviews with professionals (Healthcare professional 4, 6, 8, 12, 14 16).

Some other interesting points, detected by the shadowing, reveal how the hospital environment and management of patient flow can affect the patient experience, in particular on the day of admission. After the administrative acceptance, one patient took the wrong elevator and did not immediately reach the indicated ward. When arriving at the entrance of the ward, he found it difficult to use the intercom. When entering the ward, he was dissatisfied with the lack of staff to welcome him. When waiting in the room for surgery he showed apprehension and he reported a desire to have more information and to have a family member nearby (Patient 9). Another patient reported having received incorrect information to reach the ward and that the hospital directional signs were too small and difficult to read (Patient 1).

During the journey it is possible to identify some key steps, though with different levels of importance from patients' and professionals' perspectives. From the patient perspective and by shadowing the journey, the day of hospitalization was the most critical, and they experienced mainly negative emotions (Patient 5, 6, 7, 8, 9,12). From the interviews with the professionals it emerges that when returning to the ward after surgery patients were calm (Healthcare professional 7, 8, 16) but in the following days, they began to experience a lack of autonomy and this could make them nervous (Healthcare professional 13). Professionals involved in the pre-hospitalization phase report that waiting in the days before hospital admission can negatively affect patient experience. Patients can feel abandoned, if no one gives them information on the outcome of the outpatient clinic examination, or if all the procedures related to hospitalization are not properly programmed (Healthcare professional 1, 9).

Through analysis of the data collected four main themes emerged underlying both the shadowing and the interviews: the information gap; the covering patient-professionals relationship; the effectiveness of family closeness; and the micro-integration of hospital services.

The most significant issues are reported below.

## The information gap

When interviewed, patients did not mention any problems with the information received in the course of their hospital journey. However, when patients were shadowed on arrival at the hospital, they did not seem to be aware of any information regarding their hospitalization (e.g. visiting hours for family members, the hospital route to the ward), but asked the first professionals they met. The patients seemed lost, especially after going through the admission process and on looking for their assigned wards. Moreover, when they arrived in the ward they needed information about their hospitalization, but healthcare professionals did not immediately assist them (Patient 5, 6, 9). This seemed to contribute to their state of anxiety about the surgery. This issue is confirmed by what the professionals reported. When they arrive at the hospital, patients put the same questions to any professional they come into contact with (Healthcare professional 4, 6). A nurse reports how each patient has *"so many anxieties, fears, uncertainties, questions, as soon as he steps into the ward and I follow him, until he leaves the ward"* (Healthcare professional 1). A head nurse reports *"Family members also ask many questions. Many times it seems that what was already explained by the doctor, actually, has not sunk in (. . .) And so here they repeat the same questions many times, in different ways. What worries them a lot (. . .) is what will happen after discharge, when 'I find him at home or in a rehabilitation clinic'"* (Healthcare professional 13). Apprehension before surgery was observed in one patient, even though the patient claimed to have received very good information on how the surgery would be performed (Patient 2).

The time of waiting while the patient is in the operating theatre seems endless for family members, and waiting without information is a cause of anxiety (Healthcare professional 6). Professionals recognize the importance of informing the patient and family members about procedures, clinical pathways and pain management, before surgery (Healthcare professional 1, 2, 4, 7).

## The covering patient-professionals relationship

The relationship between patient and professionals is a key issue for the quality of the service perceived by the patient, even when the health care provider fails to respond immediately to the patient's needs. Indeed, as many as 35 touchpoints occur throughout the patient journey (Table 3). What the patient thinks and feels on this topic, emerges especially from the interviews, while the shadowing is not able to immediately capture thoughts or observations re-elaborated by the patient. In particular, when interviewed the patients remember, even after some time, some aspects of the relationship with professionals that are not directly related to clinical care, but which are perceived as being of value for the patients, since in these they receive attention as an individual. Even after some time, a patient remembered: *"Early in the morning the nurse came to say goodbye before she went off duty, because I was being discharged later that day, so she wouldn't see me again. Really good."* (Patient 3). A patient also remembered a rough response to a request for help to get dressed after the X-ray during outpatient clinic examination (Patient 2). Moreover, a patient pointed out how reassuring the relationship with the surgeon could be just before the surgery (Patient 3). One of the key moments appears to be the contact with the anaesthesiologist and the surgeon while the patient is waiting in the operating theatre: *"Then the anaesthesiologist told me 'Don't worry, my dear, we do*

*the epidural, we will sedate you'"* (Patient 1). From the professionals' perspective, the relationship with the patient is a key point to "buffer" a series of disruptions in the hospital journey and to reassure the patient: "*Patients always thank us because even if there is a gap in the organization and the patient has to wait a little, we apologize in the best way, with a smile"* (Healthcare professional 3). As the nurse is the first person patients encounter when entering the ward, she knows she has the important role of reassuring patients by explaining to them how to orientate themselves in the ward and which procedures will be carried out, even if patients should already have been informed about all these things (Healthcare professional 4). Professionals recognize the importance of calming patients through interaction with the surgeon especially when they are waiting just before surgery (Healthcare professional 3, 13). An orthopaedist reports, "*When you check or welcome the patient in the operating surgery where the surgeon and the anaesthesiologist are, the patient sees them and this helps him or her a lot, and so one thing that I think is in our favour (. . .) is communication, the possibility of having a point of reference"* (Healthcare professional 3).

## The effectiveness of family closeness

Family closeness is felt to be important for both patients and professionals, if programmed at the right times of the clinical journey. From the patient interviews and from shadowing it emerges that patients like family members to stay with them when waiting for surgery (Patient 4, 9). Once the surgery has been performed, when fears are diminished and pain is controlled, patients do not consider the presence of family members necessary, in particular immediately after returning to the ward from the operating theatre (Patient 1, 4). From the shadowing it emerges that after the first few days, when patients have recovered from the post-operative stage and close assistance has diminished, they then like to be with their family without interruptions for clinical-assistance reasons (Patient 10).

For professionals, family presence is important especially shortly before and after surgery, to reassure family members that the patient is doing well (Healthcare professional 5, 7, 13). When possible, professionals try to facilitate this, even outside regular visiting hours (Healthcare professional 13). In the days after surgery, "*It is mainly relatives who come from outside the city who logically stay here, maybe in a hotel or some bed & breakfast, and would like to stay in the room all day; because they say–quite rightly, as I realize–: 'But I have nothing else to do; my husband, my wife, my son is there. I'm with him'"* (Healthcare professional 12). At this stage of the clinical journey, professionals do not see the closeness of family members as a need of the patient. Immediately after surgery, patients prefer to rest rather than having many people in their room. Conversely, the presence and closeness of family can greatly affect the patient experience in the rehabilitation period, especially when it comes to discharging elderly patients (Healthcare professional 15).

## The micro-integration of hospital services

Even for a relatively simple routine surgical pathway, patients go through multiple stages. The behind-the-scenes coordination remains invisible to them and they are able to capture only some of the effects related to it. By contrast, professionals emphasize many critical issues in the management of the patient journey that affect the patient's experience.

When interviewed, patients reported the difficulty of having to move from one clinic to another during the outpatient clinic examination (Patient 1). Before hospital admission, an admissions office administrator shows how necessary it is to "*decrease calls to the patient (. . .), also depending on their age which is on average quite advanced . . .. cut out some calls that often from their point of view are unconnected. For instance, on one day I call you for admission, then*

*the doctor calls you for blood tests, then another doctor calls you to arrange the meeting (. . .), then if you take cardioaspirin the doctor calls you to give you information on cardioaspirin . . . All these calls could be grouped into maybe one by the doctor and one by the administrative staff"* (Healthcare professional 10). At the time of hospital admission some critical points are revealed by the shadowing. After arriving at the hospital, patients waited an average of 21 minutes before being taken in charge by the Administrative Office to carry out admission procedures. At the end of the administrative registration procedures, patients made their own way to the ward, taking an average of 11 minutes. In this time, patients could get lost; they experienced anxiety about not getting to the right place, and waited outside the closed door of the ward without knowing what to do (Patient 5, 6, 7, 9). During their hospitalization, patients reported a lack of communication: a drug intolerance reported in their previous admissions had not been recorded in the notes. Orthopaedists reported critical issues concerning the management of operating theatres, such as delays in transporting patients from the ward to the operating theatre or delays in preparing the operating theatre for the next operation (Healthcare professional 2, 3). The accumulation of such delays could lead to the cancellation of the last scheduled patient, with a negative impact on the patient who had been waiting in a state of anxiety for many hours (Healthcare professional 2). A head nurse reported that waiting for transport to and from the radiology department for the post-operative radiography could slow down all the care processes, make the patient wait unnecessarily, and increase the pain, due to the temporary suspension of the continuous-infusion pain-killer (Healthcare professional 12). Finally, a patient reported that she was offered no choice when she was transferred to the rehabilitation unit recommended by the doctors, and she expressed the desire for a follow-up visit by the same doctor who had operated on her (Patient 1).

## Discussion

Exploring the individual patient journey can lead healthcare organizations to improve patient experience by focusing on the patient perspective, rather than the provider perspective [39]. Understanding what organizations can do to improve patient experience is critical [40]. However, the literature is still exploring the best methods to capture the patient's experience [17, 23, 30]. This study deals with the lived experience of orthopaedic patients by capturing the different points of view of patients and professionals on individual hospital patient journeys. Patients' reported experience is analysed by shadowing them during hospitalization and by interviewing them at the end of the whole journey.

Historically, researchers and health care managers have focused on the study of how to achieve effective care through the definition of clinical pathways and by increasing patient adherence to treatment. However, reducing the patient's path to the clinical perspective may fail to reveal aspects that are relevant to patients, that influence their experience and their perception of quality of service [42, 42]. In this study on patients' hospital journeys, some important issues emerged through the shadowing of the hospital journey of the patients, and interviews with the key players. With the integrated use of these methods it was possible to identify which touchpoints are most critical for the patient, when family closeness is most effective, and how professionals can provide for the needs shown by patients over the entire journey. If on one hand the study of clinical pathways is now heading towards the active involvement of patients in decisions related to their own health issues [43], on the other hand the analysis of the hospital journey from a patient perspective can lead organizations to improve cross-hospital processes by creating procedures and focusing healthcare professionals on overall patient experience.

In line with Liberati's analysis [30], the shadowing method can contribute to patient-centredness by considering all the aspects of service delivery, not just the clinical one. In this study, both interviews and shadowing are able to "see the world from someone else's point of view" [24]. However, the patient's observations, focused on the whole service experienced, can reveal areas of potential improvement of the patient experience not otherwise identifiable. Shadowing highlights what the patient experiences in the different contexts and when going through one service and on to another, which professionals do not see since this falls outside the scope of their direct responsibility. Moreover, unlike using diaries completed by patients [23], this methodology allows the patient to be observed in the moment and in the spaces in which the relationship with the professionals takes place. However, this necessarily determines a subjective interpretation of what the researcher observes with respect to what the patient affirms.

Unlike what was pointed out by Gill [44], when dealing with the patient journey perspective, shadowing has an important potential for revealing invisible steps and spaces of the journey, more than intimate spaces and micro-processes of the decision. It is true that even now, in the healthcare sector, the provider establishes the patient path, while the patient is 'carried forward' through processes designed and managed by others.

In this study, when interviewed after time, patients focused on the overall clinical experience, forgetting other issues related to their hospitalization. For example, when interviewed, patients reported that they had had all the information they needed, while when shadowed shortly before the surgery the same patients appeared lost and asked for information from all the professionals they met. These data are also confirmed by interviews with professionals, who reported how highly emotional touchpoints, such as telling the patient they needed an operation, or the time immediately before transfer to the operating theatre, may affect patients' perceptions and the effectiveness of the information [41, 45]. As suggested by Ziebland, there is a difference between what patients said they experienced and what they actually experienced in real-life settings [20]. In this sense, the use of shadowing helps to understand the experience in a real time context. Moreover, it is always useful to evaluate whether the tools and information methods used for giving information to patients are effective, and which is the best moment for each patient to receive all the information they need, by considering their ability to absorb the information in a stressful situation [46–48].

In this study, both patients and professionals recognized the value of a personalized relationship in improving patient experience. Moreover, professionals report how a good relationship with the patient can compensate for the organization's inefficiencies. Interaction with the patient is especially important in the perceived patient-critical touchpoints. However, relevant steps of the journey are different from patients' and from professionals' points of view. From the patient's point of view, the most critical steps occur when entering the hospital and just before surgery, where their emotional involvement is greater. On the other hand, from the professionals' point of view, planning hospitalization and preparing patients for surgery is one of the most critical steps that affect patient experience. Indeed, patients, when interviewed, seem not to perceive critical issues in what happens 'behind the scenes', while professionals are able to identify issues related to the organization that can positively or negatively affect patients' experience. These results highlight how frontline professionals are the key players in transforming organizational procedures into personalized care pathways, but the misalignment of views should be considered when improving the hospital journey by including the patients' perspective.

The study has important limitations with respect to the sample and the setting considered and therefore its potential for generalization may be limited. The issues that emerged would need to be studied in depth in different care settings and with other types of patients to allow comparison of data and methodologies.

## Conclusions

Patients' experiences have become increasingly central to assess the performance of healthcare organizations and to redesign the services around the real needs of patients [20, 41, 42, 45, 49]. In this study, the analysis of the hospital journey from the patient perspective and the integration of three different standpoints, patient shadowing, healthcare professional interviews and patient interviews, highlights important areas of improvement otherwise hidden by the analysis of the clinical pathway only.

The nature of the study and its originality by subject matter and methods adopted can stimulate both academics and healthcare managers to explore important new fields. On the one hand, it is important to further investigate methodologies for capturing the patient experience and use it deeply and effectively at various organizational levels. In this way, shadowing seems to give a more patient-centric perspective, but it raises questions about its effectiveness as a single methodology for gathering the whole patient experience within a complex hospital process. On the other hand, the results of this study are a starting-point for healthcare managers who want to improve a key cross-functional hospital process in which the patient is the main actor. By considering the overall patient experience, as well as services performance and clinical pathways, they will able to create a distinctive value both for the patient and for the organization.

## Acknowledgments

We are grateful to Eugenia Di Sabatino and Michela Ceri for their contribution to data collection.

## Author Contributions

**Conceptualization:** Raffaella Gualandi, Cristina Masella, Daniela Tartaglini.

**Data curation:** Raffaella Gualandi, Cristina Masella, Daniela Viglione.

**Formal analysis:** Raffaella Gualandi, Cristina Masella, Daniela Viglione.

**Investigation:** Raffaella Gualandi, Cristina Masella, Daniela Viglione.

**Methodology:** Raffaella Gualandi, Cristina Masella.

**Project administration:** Raffaella Gualandi, Daniela Viglione.

**Supervision:** Daniela Tartaglini.

**Writing – original draft:** Raffaella Gualandi, Cristina Masella.

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
