## [Decision Letter · Decision Letter 0]

30 Aug 2019

PONE-D-19-19443

Exploring the hospital patient journey: how can we capture the patient’s experience?

PLOS ONE

Dear Dr. Gualandi ,

Thank you for submitting your manuscript to PLOS ONE. After careful consideration, we feel that it has merit but does not fully meet PLOS ONE’s publication criteria as it currently stands. Therefore, we invite you to submit a revised version of the manuscript that addresses the points raised during the review process.

Please address reviewer comments as well as the following:

Literature review - further detail on the theoretical basis of shadowing and rationale for it's use.

Methods

Data collection - was there any prior relationship between the researcher and potential participants?  How were issues of possible coercion dealt with?

Please provide further information on shadowing and how the field notes were dealt with

Much greater detail is required in reference to Plutchik's Wheel of Emotions and the rationale for its use as a framework in this study.

Line 146 include "with participant permission"

Results

Please include some minimal demographic characteristics of participants.

Please provide a better link between the text and Table 2

The data presented seems to come from the interviews - where is the data from the shadowing included?

We would appreciate receiving your revised manuscript by 30 September 2019. To enhance the reproducibility of your results, we recommend that if applicable you deposit your laboratory protocols in protocols.io, where a protocol can be assigned its own identifier (DOI) such that it can be cited independently in the future. For instructions see: http://journals.plos.org/plosone/s/submission-guidelines#loc-laboratory-protocols

We look forward to receiving your revised manuscript.

Kind regards,

Rosemary Frey

Academic Editor

PLOS ONE

Journal Requirements:

3. Please consider including more information on the number of interviewers, their training and characteristics.

NO - The funders had no role in study design, data collection and analysis, decision to publish, or preparation of the manuscript

a) Please provide an amended Funding Statement that declares *all* the funding or sources of support received during this specific study (whether external or internal to your organization) as detailed online in our guide for authors at http://journals.plos.org/plosone/s/submit-now.  

b) Please state what role the funders took in the study.  If any authors received a salary from any of your funders, please state which authors and which funder. If the funders had no role, please state: "The funders had no role in study design, data collection and analysis, decision to publish, or preparation of the manuscript."

Reviewers' comments:

Reviewer's Responses to Questions

**Comments to the Author**

1. Is the manuscript technically sound, and do the data support the conclusions?

Reviewer #1: Partly

Reviewer #2: Yes

2. Has the statistical analysis been performed appropriately and rigorously? 

Reviewer #1: N/A

Reviewer #2: Yes

3. Have the authors made all data underlying the findings in their manuscript fully available?

Reviewer #1: Yes

Reviewer #2: No

4. Is the manuscript presented in an intelligible fashion and written in standard English?

Reviewer #1: No

Reviewer #2: Yes

5. Review Comments to the Author

Reviewer #1: Title: Exploring the hospital patient journal: how can we capture the patient’s experience?

Manuscript number: PONE D 19 19443

General comments

Your title is Exploring the patient journey: how can we capture the patient experience?

And the research questions are Which aspects of hospital patient journey experience may be captured by 3 different standpoints – shadowing, professional and patient interviews. What does the patient experience through the hospital journey and how can it be captured?

The complexity and number of steps in the patient journey decoded by your research is very interesting as were other findings of patient journey. I commend you for using an ethnographic approach to try to capture the lived experiences of the patients your followed.

These tools captured patient journey, the physical aspects of movement through a hospital experience; however, I am interested to know other aspects of patient experience. For example, did the hospital context/system facilitate patient understanding of their own condition and care (health literacy)? Could that be a lens you view the data gleaned from shadowing?

Shadowing as a research methodology also requires more analysis in the paper. I don’t get a sense of the ‘rich’ description that the ethnographic shadowing approach should generate. It would be good to see what else that data revealed beside the annotation of components of the hospital journey, given you were trying to capture the broader experience of patients – viz Line 328 - patient mainly feels and remembers, while shadowing highlights what the patient experiences in the different contexts and going through one service and on to another, which professionals do not see since this falls outside the scope of their direct responsibility.

See also Gill, R. (2011). The shadow in organizational ethnography: moving beyond shadowing to spect -acting. Qualitative Research in Organizations and Management: An International Journal, 6(2), 115-133

Line 15 – delete “the’ – definite article.

Line 26 – “professionals” – a possessive?

Line 32 – comma after process

Line 43 – vague reference to “some others”

Line 87 – reference for phenomenological-hermeneutic approach

Line 90 – unless that is the name of the hospital, only ‘Italian’ should have capitalisation; however, your use of the indefinite article makes me think it is not a proper noun.

Use of appendix – perhaps not needed – the table could be condensed and summarised inside a table in the text or using words.

Lines 119-122 This is unclear - we need some researcher reflexivity here – what were the researchers’ roles in the delivery of patient care?

Line 136 – patient’s journey/experience or patients’ journeys/experiences?

Line 140 – can you explain more about sampling decisions for professional staff, esp the nurses

Line 157 – Plutchik’s Wheels of Emotion needs some backgrounding – why is that appropriate and how did it fit with your phenomenological-hermeneutic approach and your research question? Table 2 reveals that patient emotions were captured, but was there any particular part of the 3 different standpoints which captured emotion?

Line 180 – rules of long quotes – over 40 words – indented? Italics?

Line 195 - “they began to experience a lack of autonomy and this could make them nervous (N2)” – Could you clarify what the coding stands for in your results? E.g. N or P or H? Also does one reference (N2) here mean only one person had this experience?

Line 210 – verb aspect – going/gone – delete ‘having’

Lines 233 on – was this data gained from the interviews or shadowing?

? hegemonic/power relationships in that context? Any critical analysis?

Line 227 – The relationship between Table 2 and patient-professional relationships could be clarified further.

Line 373/4 – where was the review of the shadowing methodology challenges?

Reviewer #2: Overall this is an interesting, well written study. The topic of the patient experience is important and new ways of capturing it, and using this information to improve patient care, are central to modern healthcare. Specific points follow.

Page 2, line 15: “hospital patient journey experience” is an ugly noun cluster, please unpack – “the experience of the hospital patient journey”?

Page 4, line 64 and in discussion: you talk a lot about different data collection methods and perspectives on the patient experience and mention a number of them – three complementary forms of which your study covers. However, one rather obvious method and perspective that you fail to mention is a hospital-stay diary completed by the patient themselves. This should at least be mentioned in discussion as another such method. A recent good example of this method is: Webster CS, Jowsey T, Lu LM, et al. Capturing the experience of the hospital-stay journey from admission to discharge using diaries completed by patients in their own words: a qualitative study. BMJ Open 2019;9:e027258. doi:10.1136/bmjopen-2018-027258

Page 4, line 70: “what he/she experiences IN real-time…”

Page 4, line 72: “accurate and real data on the experience” – what does “real data” mean in this context? Who’s definition of real are you using? Better to use a different word here I think.

Page 5, line 92: Line starting “Scheduled surgical patients were…” – this sentence doesn’t make much sense to me.

Page 6, line 108: Sentence starting “Inclusion criteria were scheduled…” – this sentence doesn’t make much sense either. I am unsure how inclusion criteria get scheduled or performed? Do you mean the inclusion criteria were that patients must be scheduled for their procedure for the shadowing phase and must have had the procedure performed for the interviews?

Page 6, line 113: “consent to participation in the study” – should be consent to participate…

Page 7, line 137: “Professionals ranged…” Please make it clear that you are talking about healthcare professionals throughout – there are other kinds of professionals.

Page 8, line 154: In my understanding content analysis and thematic analysis are not the same things – yet here you appear to suggest that they are? It looks more to me that you did a thematic analysis.

Page 8, line 160 (table 1): I am unconvinced that the unit of meaning is a good one to support the significance of the text, or the theme. “I didn’t understand anything”, to me does not demonstrate calming of the patient, or covering professional relationships – I am sure the patient would be much calmer if they did understand what was going on, and this is the anaesthetist’s professional obligation.

Page 9, line 173: “each actor allowed…” who is the actor here? The patients, clinician or researcher?

Page 9, line 186 and elsewhere: I can guess that P1 means patient one (but please define), but what does PJ1, H1 etc mean? Please define on first use of each numbering scheme.

Page 10, line 219: If patients ask the same questions over and over why does the hospital not supply them with a simple written information pack with frequently asked questions (FAQs)?

Page 12, table 2: What is the significance of the filled dots vs the unfilled dots? Please explain or make consistent.

Page 13, 241: we see the same quote as from Table 1 in the text here, why? Why not use another? Also it also seems odd to me that the anaesthetist would not address the patient by her name, rather than calling her “lady”, which actually seems rather rude.

Page 15, line 277: “simple ordinary surgical pathway…” Ordinary in this context sounds strange, do you mean routine?

Page 15, line 296: “had not been passed on.” – or recorded in the notes?

Page 17, line 324: “…perceptions and effectiveness of the information” – surely it is more about the patient’s ability to absorb the information in a stressful situation rather than the effectiveness of the information itself?

Page 18, line 359: “does not allow any generalization of results” – I would say that generalisation maybe limited rather than entirely ruled out. For example, your findings are substantially similar to those in the BMJ Open paper I mentioned previously using the patient diary method.

END

6. PLOS authors have the option to publish the peer review history of their article (what does this mean?). If published, this will include your full peer review and any attached files.

Reviewer #1: No

Reviewer #2: No

---

## [Author Response · Author response to Decision Letter 0]

5 Oct 2019

AUTHORS’ RESPONSES TO REFEREES’ AND EDITOR’S COMMENTS

• Literature review - further detail on the theoretical basis of shadowing and rationale for it's use.

We have included more details in the introduction to explain the theoretical basis of shadowing and the rational for its use.

• Methods

Data collection - was there any prior relationship between the researcher and potential participants? How were issues of possible coercion dealt with?

We have added this information in the first paragraph: “There were no prior relationships between researchers and patients; while two researchers knew some healthcare professionals because they worked in the same hospital, though in different units and without patient care roles. The possibility of coercion was minimized by guaranteeing data anonymity, by requesting voluntary participation in the study and by dealing with issues on which the researchers had no power to influence anything or anyone at hospital managerial level.”

Please provide further information on shadowing and how the field notes were dealt with.

We have provided further information on shadowing by explaining in more detail how we collected data.

Much greater detail is required in reference to Plutchik's Wheel of Emotions and the rationale for its use as a framework in this study.

We have added more details on Plutchik's Model and to its use in our study, in the ‘Data Collection’ section.

Line 146 include "with participant permission"

Added.

• Results

 Please include some minimal demographic characteristics of participants.

We have included the main demographic characteristics by summarizing them in a paragraph at the beginning of the ‘Results’ section and we have transferred the table from the Appendix to the main manuscript, naming it Table 2.

Please provide a better link between the text and Table 2

The data presented seems to come from the interviews - where is the data from the shadowing included?

We have renamed Table 2 as Table 3 and we have added some data in the 'The hospital patient journey' Section to better link the text to the results shown in the table. In this way, data deriving from shadowing are better represented.

Reviewer's Responses to Questions

Comments to the Author

1. Is the manuscript technically sound, and do the data support the conclusions?

 Reviewer #1: Partly

 Reviewer #2: Yes

The manuscript has been significantly revised in light of your suggestions in order to make its validity clearer.

2. Has the statistical analysis been performed appropriately and rigorously? 

 Reviewer #1: N/A

 Reviewer #2: Yes

3. Have the authors made all data underlying the findings in their manuscript fully available?

Reviewer #1: Yes

Reviewer #2: No

In accordance with the qualitative nature of the study, all data are now included as part of the main manuscript.

 4. Is the manuscript presented in an intelligible fashion and written in standard English?

Reviewer #1: No

Reviewer #2: Yes

We have had external proof reading of the manuscript carried out by a native English translator.

5. Review Comments to the Author

Reviewer #1: Title: Exploring the hospital patient journal: how can we capture the patient’s experience?

Manuscript number: PONE D 19 19443

General comments

• Your title is Exploring the patient journey: how can we capture the patient experience?

And the research questions are Which aspects of hospital patient journey experience may be captured by 3 different standpoints – shadowing, professional and patient interviews. What does the patient experience through the hospital journey and how can it be captured?

We have rewritten the title so that it is more consistent with the research questions. The new title is “Exploring the hospital patient journey: what does the patient experience ?”

• The complexity and number of steps in the patient journey decoded by your research is very interesting as were other findings of patient journey. I commend you for using an ethnographic approach to try to capture the lived experiences of the patients your followed. These tools captured patient journey, the physical aspects of movement through a hospital experience; however, I am interested to know other aspects of patient experience. For example, did the hospital context/system facilitate patient understanding of their own condition and care (health literacy)? Could that be a lens you view the data gleaned from shadowing? Shadowing as a research methodology also requires more analysis in the paper. I don’t get a sense of the ‘rich’ description that the ethnographic shadowing approach should generate. It would be good to see what else that data revealed beside the annotation of components of the hospital journey, given you were trying to capture the broader experience of patients – viz Line 328 - patient mainly feels and remembers, while shadowing highlights what the patient experiences in the different contexts and going through one service and on to another, which professionals do not see since this falls outside the scope of their direct responsibility.

See also Gill, R. (2011). The shadow in organizational ethnography: moving beyond shadowing to spect -acting. Qualitative Research in Organizations and Management: An International Journal, 6(2), 115-133

We have revised the whole paper in order to give greater value to shadowing and to describe the results obtained from this methodology. At the same time, we have tried to maintain the paper's goal of comparing different methodologies rather than focusing on the effectiveness of shadowing.

Specifically, we added the following points:

- as requested in the previous comments, in the ‘Result’ Section we have better distinguished the data coming from the shadowing from those from the interviews;

- we have included a paragraph in the "The information gap" Section with data collected from the hospital context and their impact on the patient's experience;

- we have included in the “Hospital patient journey” Section further data on what was observed by shadowing patients;

- we have included in the discussion considerations on the potential and limits of shadowing methodology when applied to the patient journey.

• Line 15 – delete “the’ – definite article.

Amended.

• Line 26 – “professionals” – a possessive?

Amended.

• Line 32 – comma after process

Amended.

• Line 43 – vague reference to “some others”

Amended.

• Line 87 – reference for phenomenological-hermeneutic approach

Added.

• Line 90 – unless that is the name of the hospital, only ‘Italian’ should have capitalisation; however, your use of the indefinite article makes me think it is not a proper noun.

Amended.

• Use of appendix – perhaps not needed – the table could be condensed and summarised inside a table in the text or using words.

We have inserted the table in the main manuscript and we have named it “Table 2”.

• Lines 119-122 This is unclear - we need some researcher reflexivity here – what were the researchers’ roles in the delivery of patient care?

As requested by the editor, we have specified the relationship between researchers and participants in the Methods section. We also have specified in the ‘Data collection’ section that students had no role in patient delivery of care. Moreover, in the discussion we have considered the subjective interpretation of the researcher in the shadowing methodology.

• Line 136 – patient’s journey/experience or patients’ journeys/experiences?

Amended.

• Line 140 – can you explain more about sampling decisions for professional staff, esp the nurses

We have specified the sampling technique. We introduced further specification in the text.

• Line 157 – Plutchik’s Wheels of Emotion needs some backgrounding – why is that appropriate and how did it fit with your phenomenological-hermeneutic approach and your research question? Table 2 reveals that patient emotions were captured, but was there any particular part of the 3 different standpoints which captured emotion?

As requested by the editor, we have added in the ‘Data Collection’ section more details in reference to Plutchik's Model and the rationale for its use in our study. In the 'The hospital patient journey' Section we have specified that the identified emotions coincide with what was detected by the shadowing and by the interviews with healthcare professionals, but they are different from what the patient reported during the interviews. In the table, renamed table 3, we have specified the column label.

• Line 180 – rules of long quotes – over 40 words – indented? Italics?

We shortened and distinguished the two quotes. We put all quotes into italics.

• Line 195 - “they began to experience a lack of autonomy and this could make them nervous (N2)” – Could you clarify what the coding stands for in your results? E.g. N or P or H? Also does one reference (N2) here mean only one person had this experience?

We have made the subject of each quotation explicit. We codified the number alone by connecting quotes with Table 2 (Subject / Number).

• Line 210 – verb aspect – going/gone – delete ‘having’

Amended.

• Lines 233 on – was this data gained from the interviews or shadowing?

 ? hegemonic/power relationships in that context? Any critical analysis?

We revised the paragraph to better define the concept and to specify the source of the data.

• Line 227 – The relationship between Table 2 and patient-professional relationships could be clarified further.

Within the first paragraph of the "The covering patient-professionals relationship" Section, we created a link with the touchpoints represented in the renamed Table 3.

• Line 373/4 – where was the review of the shadowing methodology challenges?

We changed the statement to achieve greater consistency with the results achieved by this study.

Reviewer #2: Overall this is an interesting, well written study. The topic of the patient experience is important and new ways of capturing it, and using this information to improve patient care, are central to modern healthcare. Specific points follow.

• Page 2, line 15: “hospital patient journey experience” is an ugly noun cluster, please unpack – “the experience of the hospital patient journey”?

Amended

• Page 4, line 64 and in discussion: you talk a lot about different data collection methods and perspectives on the patient experience and mention a number of them – three complementary forms of which your study covers. However, one rather obvious method and perspective that you fail to mention is a hospital-stay diary completed by the patient themselves. This should at least be mentioned in discussion as another such method. A recent good example of this method is: Webster CS, Jowsey T, Lu LM, et al. Capturing the experience of the hospital-stay journey from admission to discharge using diaries completed by patients in their own words: a qualitative study. BMJ Open 2019;9:e027258. doi:10.1136/bmjopen-2018-027258

We have carefully considered the suggested study and we have updated our analysis by commenting on it in the introduction and citing it in the discussion. 

• Page 4, line 70: “what he/she experiences IN real-time…”

Amended.

• Page 4, line 72: “accurate and real data on the experience” – what does “real data” mean in this context? Who’s definition of real are you using? Better to use a different word here I think.

Amended.

• Page 5, line 92: Line starting “Scheduled surgical patients were…” – this sentence doesn’t make much sense to me.

We reframed the sentence.

• Page 6, line 108: Sentence starting “Inclusion criteria were scheduled…” – this sentence doesn’t make much sense either. I am unsure how inclusion criteria get scheduled or performed? Do you mean the inclusion criteria were that patients must be scheduled for their procedure for the shadowing phase and must have had the procedure performed for the interviews?

We have clarified the inclusion criteria in the text.

• Page 6, line 113: “consent to participation in the study” – should be consent to participate…

Amended.

• Page 7, line 137: “Professionals ranged…” Please make it clear that you are talking about healthcare professionals throughout – there are other kinds of professionals.

Amended.

• Page 8, line 154: In my understanding content analysis and thematic analysis are not the same things – yet here you appear to suggest that they are? It looks more to me that you did a thematic analysis.

Amended. We conducted a thematic analysis.

• Page 8, line 160 (table 1): I am unconvinced that the unit of meaning is a good one to support the significance of the text, or the theme. “I didn’t understand anything”, to me does not demonstrate calming of the patient, or covering professional relationships – I am sure the patient would be much calmer if they did understand what was going on, and this is the anaesthetist’s professional obligation.

The literal translation of the text from Italian may have misrepresented the meaning of what we wanted to report. We have changed the example to report a clearer quotation.

• Page 9, line 173: “each actor allowed…” who is the actor here? The patients, clinician or researcher?

We have specified.

• Page 9, line 186 and elsewhere: I can guess that P1 means patient one (but please define), but what does PJ1, H1 etc mean? Please define on first use of each numbering scheme.

We have made the subject of each quotation explicit. We codified the number alone by connecting quotes with Table 2 (Subject / Number).

• Page 10, line 219: If patients ask the same questions over and over why does the hospital not supply them with a simple written information pack with frequently asked questions (FAQs)?

Even though it does not emerge from the interviews, the hospital already uses written information material. In our work we want to emphasize that, regardless of the patient's tools and information methods, it is always useful to evaluate whether these are effective and which is the best moment for the patient to receive all the information they need. We made this concept explicit in the discussion.

• Page 12, table 2: What is the significance of the filled dots vs the unfilled dots? Please explain or make consistent.

Amended.

• Page 13, 241: we see the same quote as from Table 1 in the text here, why? Why not use another? Also it also seems odd to me that the anaesthetist would not address the patient by her name, rather than calling her “lady”, which actually seems rather rude.

We changed the quote in the Table 1. We have replaced the word “lady” with “my dear” to better convey the note of kindness that is meant in the Italian language “Signora”.

• Page 15, line 277: “simple ordinary surgical pathway…” Ordinary in this context sounds strange, do you mean routine?

Amended.

• Page 15, line 296: “had not been passed on.” – or recorded in the notes?

Amended.

• Page 17, line 324: “…perceptions and effectiveness of the information” – surely it is more about the patient’s ability to absorb the information in a stressful situation rather than the effectiveness of the information itself?

We have specified in the lines below.

• Page 18, line 359: “does not allow any generalization of results” – I would say that generalisation may be limited rather than entirely ruled out. For example, your findings are substantially similar to those in the BMJ Open paper I mentioned previously using the patient diary method.

Amended.

 END

---

## [Decision Letter · Decision Letter 1]

24 Oct 2019

Exploring the hospital patient journey: what does the patient experience?

PONE-D-19-19443R1

Dear Dr. Gualandi,

We are pleased to inform you that your manuscript has been judged scientifically suitable for publication and will be formally accepted for publication once it complies with all outstanding technical requirements.

With kind regards,

Rosemary Frey

Academic Editor

PLOS ONE

Additional Editor Comments (optional):

Reviewers' comments:

Reviewer's Responses to Questions

**Comments to the Author**

1. If the authors have adequately addressed your comments raised in a previous round of review and you feel that this manuscript is now acceptable for publication, you may indicate that here to bypass the “Comments to the Author” section, enter your conflict of interest statement in the “Confidential to Editor” section, and submit your "Accept" recommendation.

Reviewer #1: All comments have been addressed

Reviewer #2: All comments have been addressed

2. Is the manuscript technically sound, and do the data support the conclusions?

Reviewer #1: Yes

Reviewer #2: Yes

3. Has the statistical analysis been performed appropriately and rigorously? 

Reviewer #1: N/A

Reviewer #2: Yes

4. Have the authors made all data underlying the findings in their manuscript fully available?

Reviewer #1: Yes

Reviewer #2: Yes

5. Is the manuscript presented in an intelligible fashion and written in standard English?

Reviewer #1: Yes

Reviewer #2: Yes

6. Review Comments to the Author

Reviewer #1: 1. Check for use of Tab instead of Table (see Lines 197 & 281)

2. Sentence (Line 397) doesn’t makes sense to me. Perhaps “In contrast to Gill (I’m still not sure though what the contrast is.)

Unlike what was pointed out by Gill [44], when dealing with the 397 patient journey perspective,

398 shadowing has an important potential for revealing invisible steps and spaces of the journey, more

399 than intimate spaces and micro-processes of the decision.

Reviewer #2: This is an excellent paper, and the authors have carefully addressed my review points.

My only very small remaining suggestion would be to use the Italian "signora" instead of "my dear" - the English translation doesn't really capture the original meaning, and signora is a well known term in English which seems more appropriate in the context of the quote.

7. PLOS authors have the option to publish the peer review history of their article (what does this mean?). If published, this will include your full peer review and any attached files.

Reviewer #1: No

Reviewer #2: No

---

## [Editor Report · Acceptance letter]

18 Nov 2019

PONE-D-19-19443R1 

Exploring the hospital patient journey: what does the patient experience? 

Dear Dr. Gualandi:

I am pleased to inform you that your manuscript has been deemed suitable for publication in PLOS ONE. Congratulations! Your manuscript is now with our production department. 

With kind regards,

on behalf of

Dr. Rosemary Frey 

Academic Editor

PLOS ONE